# Effects of *Lactobacillus plantarum* HW1 on Growth Performance, Intestinal Immune Response, Barrier Function, and Cecal Microflora of Broilers with Necrotic Enteritis

**DOI:** 10.3390/ani13243810

**Published:** 2023-12-10

**Authors:** Peng Chen, Huimin Lv, Weiyong Liu, Yang Wang, Kai Zhang, Chuanyan Che, Jinshan Zhao, Huawei Liu

**Affiliations:** 1College of Animal Science and Technology, Qingdao Agricultural University, Qingdao 266109, China; 2College of Animal Science, Anhui Science and Technology University, Fengyang 233100, China

**Keywords:** *Lactobacillus plantarum*, broiler, necrotic enteritis, immunity response, intestinal barrier function

## Abstract

**Simple Summary:**

Necrotic enteritis leads to decreased productivity and increased mortality in broilers. Antibiotics have been used to prevent and treat necrotic enteritis. However, the overuse of antibiotics leads to drug residues and resistance. Therefore, there is an urgent need to search for a safe, green, and pollution-free feed additive. *Lactobacillus plantarum* has been reported to improve immunity and intestinal flora to enhance the intestinal health and growth performance of broilers. In our study, dietary 4 × 10^6^ CFU/g *Lactobacillus plantarum* HW1 reduced the levels of intestinal inflammatory factors and improved intestinal barrier function and cecal microflora, thus improved growth performance in broilers with necrotic enteritis. Our results provide a theoretical basis for application of *Lactobacillus plantarum* HW1 as a feed additive in broiler production.

**Abstract:**

The purpose of the study was to investigate the effects of *Lactobacillus plantarum* HW1 on growth performance, intestinal immune response, barrier function, and cecal microflora of broilers with necrotic enteritis. In total, 180 one-day-old male Cobb 500 broilers were randomly allocated into three groups comprising a non-infected control (NC) group, basal diet + necrotic enteritis challenge (NE) group, and basal diet + 4 × 10^6^ CFU/g *Lactobacillus plantarum* HW1 + necrotic enteritis challenge (HW1) group. Broilers in the NE and HW1 groups were orally given sporulated coccidian oocysts at day 14 and *Clostridium perfringens* from days 19 to 21. The results showed that the HW1 treatment increased (*p* < 0.05) the average daily gain of broilers from days 15 to 28 and from days 0 to 28 compared with the NE group. Moreover, the HW1 treatment decreased (*p* < 0.05) the oocysts per gram of excreta, intestinal lesion scores, ileal interleukin (IL) 1β and tumor necrosis factor α levels, and serum *D*-lactic acid and diamine oxidase levels, while increasing (*p* < 0.05) the ileal IL-10 level, thymus index, and protein expressions of ileal occludin and ZO-1. Additionally, the HW1 treatment decreased (*p* < 0.05) the jejunal and ileal villus height, jejunal villus height/crypt depth value, and cecal harmful bacterial counts (*Clostridium perfringens*, *Salmonella*, *Escherichia coli*, and *Staphylococcus aureus*), and increased (*p* < 0.05) the cecal *Lactobacillus* count. In conclusion, dietary supplementation with 4 × 10^6^ CFU/g *Lactobacillus plantarum* HW1 could relieve necrotic enteritis infection-induced intestinal injury and improve growth performance in broilers by improving intestinal barrier function and regulating intestinal microbiology.

## 1. Introduction

Necrotic enteritis is a common intestinal disease mainly caused by NetB-producing strains of *Clostridium perfringens* [1,2], and often associated with one or more predisposing factors, such as pathogens (e.g., *Eimeria*) and diet ingredients (e.g., fish meal) [3,4]. The toxin secreted by *Clostridium perfringens* causes necrosis of intestinal mucosal cells [5] and the toxin penetrates the mucosal barrier into the blood, causing systemic symptoms [6,7]. In addition, necrotic enteritis was reported to cause intestinal ulcers, erosions, and other adverse effects [8], leading to decreased productivity and increased mortality in broilers [9,10]. Antibiotics have been used to prevent and treat necrotic enteritis. However, the overuse of antibiotics leads to drug residues and resistance. Thus, there is an urgent need to search for a safe, green, and pollution-free feed additive.

Probiotics, as a kind of green and pollution-free active microorganism, can effectively prevent the overgrowth of pathogens and the development of diseases in the gut of broilers [11,12]. *Lactobacillus plantarum*, as a probiotic, produces organic acids during metabolism such as lactic acid, acetic acid, and propionic acid, which can lower the pH of the intestine and inhibit the proliferation of harmful bacteria [13,14]. Studies showed that dietary supplementation with *Lactobacillus plantarum* decreased the oocyst shedding rate and improved the intestinal barrier in *Eimeria*-infected broilers [15], and improved the intestinal inflammation and decreased the enteric pathogens (*Escherichia coli*, *Bacteroides fragilis*) in *Clostridium perfringens*-infected broilers [16]. Our previous works also revealed that *Lactobacillus plantarum* HW1, screened from the intestinal mucosa of healthy broilers, improved the growth performance and meat quality in meat geese [17], and had anti-*Clostridium perfringens* activities in vitro (unpublished data). However, the effects of *Lactobacillus plantarum* HW1 on the growth performance and intestinal health of broilers with necrotic enteritis need to be further determined. Thus, the purpose of this study was to investigate the effects of *Lactobacillus plantarum* HW1 on the growth performance, intestinal immune response, barrier function, and cecal microflora of broilers with necrotic enteritis.

## 2. Materials and Methods

### 2.1. Ethical Approval

All procedures were approved by the Animal Care and Use Committee of Qingdao Agricultural University.

### 2.2. Materials

*Lactobacillus plantarum* HW1 was isolated from intestinal mucosa of healthy broilers according to the method of Noohi et al. [18]. Briefly, the continuously diluted intestinal mucosa samples were plated on Man Rogosa and Sharpe (MRS) agar medium at 37 °C with microaerobic conditions for 48 h and milky white colonies were selected for further purification. And *Lactobacillus plantarum* HW1 was deposited in the China General Microbiological Culture Collection Center (Beijing, China; CGMCC No. 26160). We purchased a NetB toxin positive type A *Clostridium perfringens* (CVCC 2030) strain from the intestine of a broiler clinically diagnosed with necrotic enteritis at the China Institute of Veterinary Drug Control (Beijing, China). The laboratory of veterinary parasitology provided four *Eimeria* species (*Eimeria necatrix*, *Eimeria tenella*, *Eimeria maxima*, and *Eimeria acervuline*) at Qingdao Agricultural University, China.

Cryopreserved *Clostridium perfringens* CVCC2030 was thawed and cultured overnight at 37 °C in cooked meat medium (Haibo Biotechnology Co., Ltd., Qingdao, China). The culture was then transferred to Trypticase Sulfite Neomycin (TSN) medium (Haibo Biotechnology Co., Ltd., Qingdao, China) by plate streaking and cultured anaerobically for 24 h, after which the single colony was then selected and cultured in cooked meat medium (Haibo Biotechnology Co., Ltd., Qingdao, China) to logarithmic growth stage. Cryopreserved *Lactobacillus plantarum* HW1 was activated and cultured in MRS medium (Haibo Biotechnology Co., Ltd., Qingdao, China) at 37 °C for 24 h. After this, *Lactobacillus plantarum* HW1 liquid solution was diluted until the concentration of bacteria reached the required concentration.

### 2.3. Animals and Experimental Design

A total of 180 one-day-old male Cobb 500 broilers (43.22 ± 1.44 g) were randomly allocated into three groups with six replicated cages of ten broilers each for a feeding period of 28 days (Table 1). The three groups included non-infected control (NC) group, basal diet + necrotic enteritis challenge (NE) group, and basal diet + 4 × 10^6^ CFU/g *Lactobacillus plantarum* HW1 + necrotic enteritis challenge (HW1) group. The dosage of *Lactobacillus plantarum* HW1 was derived from a previous study in meat geese [17]. The HW1 group was given a diet containing 4 × 10^6^ CFU/g *Lactobacillus plantarum* HW1 daily during days 0–28. The liquid *Lactobacillus plantarum* HW1 were sprayed on the feed during mixing. The viable bacteria count of *Lactobacillus plantarum* HW1 was determined to ensure that each batch of mixed feed provided 4 × 10^6^ CFU/g of diet. The dose of oocysts of *Eimeria* spp. and *Clostridium perfringens* CVCC 2030 in necrotic enteritis model were derived from the method of Pham et al. [19]. On day 14, broilers in the NE and HW1 groups were orally given with 1 mL saline containing 4 × 10^4^ sporulated oocysts (*Eimeria necatrix* 1 × 10^4^, *Eimeria tenella* 1 × 10^4^, *Eimeria maxima* 1 × 10^4^, and *Eimeria acervuline* 1 × 10^4^), while broilers in the NC group were orally given an equivalent amount of normal saline. From days 19 to 21, all broilers in the NE and HW1 groups were daily given orally 1 mL *Clostridium perfringens* CVCC 2030 at 4 × 10^8^ CFU/mL, and the broilers in the NC group were orally given an equivalent amount of normal sterile medium. The basal diet was formulated according to the requirements of the Cobb Broiler Management Guide [20] (Table 2). The size of cages used was 70 cm × 70 cm × 40 cm (length × width × height). The lighting schedule throughout the experiment consisted of 23 h of light and 1 h of darkness.

### 2.4. Growth Performance

Feed intake was weighed daily on a replicate basis every day. After 12 h of feed withdrawal, body weight was weighed on a replicate basis at days 0, 14, and 28. Average daily feed intake (ADFI), average daily gain (ADG), and feed/gain (F/G) ratio were calculated.

### 2.5. Sample Collection

On day 28, one broiler from each replicate (6 broilers per group) was randomly selected and blood samples were collected from the jugular vein into vacuum tubes containing coagulant (silicon dioxide). Serum samples were collected by centrifugation (3000× *g*) for 10 min at 4 °C and stored in 1.5 mL Eppendorf tubes at −20 °C. Then, selected broiler killed by cervical dislocation. Spleen, thymus, and bursa of Fabricius were collected and weighted. Segments of jejunum and ileum were collected and stored in 4% buffered formaldehyde. Duodenum, jejunum, and ileum samples were collected for lesion analysis. Ileal mucosa and cecal contents were collected for further analysis.

### 2.6. Measurement of Intestinal Lesion Scores and Eimeria Oocyst Counts

On day 19, broiler excreta samples on a replicate basis were collected and the oocysts per gram of excreta (OPG) count was calculated according to the method of Youn et al. [21]. On day 28, the lesions of duodenum, jejunum, and ileum of broilers were scored by visual method according to the method of Johnson et al. [22], a score of 0 (no gross lesions), 1 (congested intestinal mucosa), 2 (small focal necrosis or ulceration; 1 to 5 foci), 3 (focal necrosis or ulceration; 6 to 15 foci), or 4 (focal necrosis or ulceration; 16 or more foci).

### 2.7. Relative Immune Organ Weight

Thymus, spleen, and bursa of Fabricius were weighed. The immune organ indices were calculated using the following formula: [immune organ weight (g)/bodyweight (g)] × 1000 [23].

### 2.8. Analysis of D-Lactic Acid, Diamine Oxidase (DAO), and Immune-Related Indices

From each replicate (6 broilers per group), 0.1 g ileum mucosa sample was homogenized with 0.9 mL normal saline. The supernatant was collected by centrifugation (3000× *g*) for 10 min. The levels of *D*-lactic acid and DAO in serum, and the levels of interleukin (IL) 1β, IL-6, IL-10, tumor necrosis factor-α (TNF-α), interferon-γ (IFN-γ), and secretory immunoglobulin A (sIgA) in the ileal mucosa were determined using ELISA kits from Jiangsu Meimian Industrial Co., Ltd. (Yancheng, China). Ileum mucosa samples were homogenized and measured for protein concentration using enhanced BCA protein assay kit (Biyuntian Biotechnology Co., Ltd., Shanghai, China), and values were expressed per milligram of protein.

### 2.9. Western Blotting Analysis

The proteins of six ileal mucosa samples from each group on day 28 were extracted using RIPA lysis buffer (Biyuntian Biotechnology Co., Ltd., Shanghai, China). Equal amounts of proteins were separated by electrophoresis, then proteins on the gel were transferred to nitrocellulose membrane. Membranes were blocked using 5% skimmed milk (Biyuntian Biotechnology Co., Ltd., Shanghai, China) and then incubated with the primary antibodies, such as anti-β-actin (Xavier Biotechnology Co., Ltd., Wuhan, China), anti-ZO-1 (Xavier Biotechnology Co., Ltd., Wuhan, China), and anti-occludin (Xavier Biotechnology Co., Ltd., Wuhan, China) overnight at 4 °C. Membranes were incubated with anti-rabbit IgG (Biyuntian Biotechnology Co., Ltd., Shanghai, China) after washing with Tris Buffered Saline Tween. The blots were then developed via a CanoScan LiDE 100 scanner (Canon, Tokyo, Japan) using the enhanced chemiluminescence kit (Biyuntian Biotechnology Co., Ltd., Shanghai, China). Quantification was carried out with Image J software (Version 1.8.0.112).

### 2.10. Intestinal Morphology

The intestinal segments were fixed in 4% formaldehyde and embedded in paraffin. Sections were placed on a glass slide and then stained with hematoxylin and eosin, and observed under a microscope (Eclipse 80i, Nikon Inc., Tokyo, Japan). Villus height and crypt depth was measured, and the villus height/crypt depth (V/C) value was calculated from above data [24]. 

### 2.11. Cultivation and Enumeration of Bacteria in Cecum

From each replicate (6 broilers per group), 1 g of cecal contents was serially diluted in 9 mL phosphate-buffered saline solution and mixed thoroughly. Total plate count was cultured with Plate Count agar (PCA) (Haibo Biotechnology Co., Ltd., Qingdao, China) under aerobic conditions at 37 °C for 48 h. *Escherichia coli* was cultured with Eosin Methylene Blue (EMB) agar (Haibo Biotechnology Co., Ltd., Qingdao, China) under aerobic conditions at 37 °C for 48 h. *Clostridium perfringens* was cultured with Tryptose Sulfite Cycloserine (TSC) agar (Haibo Biotechnology Co., Ltd., Qingdao, China) under anaerobic cabinet at 37 °C for 24 h. *Lactobacillus* was cultured with MRS agar (Haibo Biotechnology Co., Ltd., Qingdao, China) under anaerobic conditions at 37 °C for 48 h. *Staphylococcus aureus* was cultured with Baird-Parker agar (Haibo Biotechnology Co., Ltd., Qingdao, China) under aerobic conditions at 37 °C for 48 h. Salmonella was cultured with Salmonella Shigella agar (Haibo Biotechnology Co., Ltd., Qingdao, China) under aerobic conditions at 37 °C for 48 h. The microflora numbers were calculated by plate count and expressed as log_10_ CFU per gram of cecum digest.

### 2.12. Statistical Analysis

All data were analyzed with single factor analysis using SPSS Statistics 26.0 (SPSS Statistics, Chicago, IL, USA). The data on growth performance, immune organ indices, immune-related indices, intestinal morphology, intestinal permeability, and cecal microflora were analyzed using one-way ANOVA. The data on mortality, *Eimeria* oocyst counts, and intestinal lesion scores were analyzed by Kruskal–Wallis test. The differences among the three groups were compared by Duncan’s multiple comparison test. Statistical significance was determined at a significance level of *p* < 0.05 for all means and comparison groups.

## 3. Results

### 3.1. Growth Performance

There were no significant differences (*p* > 0.05) in the ADG, ADFI, and F/G ratio among the three groups from days 0 to 14 (Table 3). During days 15 to 28, the NE induction significantly decreased the ADG (*p* = 0.026) and ADFI (*p* = 0.035) compared with the NC group. Compared with the NE group, the HW1 treatment increased the ADG (*p* = 0.026). From days 0 to 28, compared with the NC group, the NE induction significantly decreased the ADG (*p* = 0.012) and ADFI (*p* = 0.013). Compared with the NE group, the HW1 treatment increased the ADG (*p* = 0.012). In addition, compared with the NC group, the NE induction significantly increased mortality (*p* = 0.031).

### 3.2. Intestinal Lesion Scores and Eimeria Oocyst Counts 

Compared with the NC group, the NE induction significantly increased the duodenal lesion score (*p* < 0.001), jejunal lesion score (*p* = 0.001), and ileal lesion score (*p* < 0.001) (Figure 1 and Table 4). Compared with the NE group, the HW1 treatment significantly decreased the duodenal lesion score (*p* < 0.001), jejunal lesion score (*p* = 0.001), and ileal lesion score (*p* < 0.001). Moreover, compared with the NC group, the NE induction significantly increased the OPG (*p* < 0.001). Compared with the NE group, the HW1 treatment significantly decreased the OPG (*p* < 0.001).

### 3.3. Immune Organ Indices

Compared with the NC group, the NE induction significantly decreased the thymus index (*p* = 0.003) and spleen index (*p* = 0.045) (Table 5). Compared with the NE group, the HW1 treatment significantly increased the thymus index (*p* = 0.003). Moreover, no significant difference (*p* > 0.05) was observed for the bursa of Fabricius index. 

### 3.4. Ileal Immune-Related Indices

Compared with the NC group, the NE induction significantly increased the TNF-α (*p* = 0.035) and IFN-γ (*p* = 0.039) levels and decreased the IL-10 (*p* = 0.015) level (Table 6). Compared with the NE group, the HW1 treatment significantly decreased the IL-1β (*p* = 0.047) and TNF-α (*p* = 0.035) levels and increased the IL-10 (*p* = 0.015) level. Meanwhile, the IL-6 and sIgA levels (*p* > 0.05) did not show significant differences among the three groups of broilers.

### 3.5. Intestinal Permeability

The serum *D*-lactic acid (*p* = 0.003) and DAO (*p* = 0.018) levels in NE group were higher than those in the NC and HW1 groups, but no significant differences were observed between the NC and HW1 groups (Figure 2). 

The ileal protein expression of occludin (*p* = 0.037) in the NE group was lower than that in the NC and HW1 groups, but no significant differences were observed between the NC and HW1 groups (Figure 3). The ileal protein expression of ZO-1 (*p* = 0.017) in the NE group was lower than that in the HW1 group, but no significant differences were observed between the NC and HW1 groups. 

### 3.6. Intestinal Morphology 

As shown in Figure 4 and Table 7, on day 28, compared with the NC group, the NE induction significantly decreased the villus height (*p* = 0.009) in the ileal and villus height (*p* = 0.008) and V/C value (*p* = 0.025) in the jejunal. Compared with the NE group, the HW1 treatment significantly increased the villus height (*p* = 0.009) in the ileal and villus height (*p* = 0.008) and V/C value (*p* = 0.025) in the jejunal.

### 3.7. Cecal Bacterial Counts

Compared with the NC group, the NE induction significantly increased the *Clostridium perfringens* (*p* < 0.001), *Salmonella* (*p* < 0.001), *Escherichia coli* (*p* = 0.001), and *Staphylococcus aureus* counts (*p* = 0.020) in the cecum, and decreased the *Lactobacillus* count (*p* < 0.001) in the cecum (Table 8). The *Clostridium perfringens* count without log_10_ transformation in the NE group was over eight times higher than that in the NC group. Compared with the NE group, the HW1 treatment significantly decreased the *Clostridium perfringens* (*p* < 0.001), *Salmonella* (*p* < 0.001), *Escherichia coli* (*p* = 0.001), and *Staphylococcus aureus* counts (*p* = 0.020) in the cecum, and increased the *Lactobacillus* count (*p* < 0.001). However, the total plate count (*p* > 0.05) did not show significant differences among the three groups of broilers.

## 4. Discussion

The protein leakage caused by coccidia enters the intestinal cavity, thereby promoting the rapid proliferation and pathogenicity of *Clostridium perfringens*, and reducing the digestion and absorption capacity of the small intestine, and thus reduces the growth performance [25]. Intestinal lesion scores are often used to evaluate the severity of necrotic enteritis [26,27]. In this study, necrotic enteritis increased the mortality, intestinal lesion scores, and OPG, and decreased the ADG and ADFI, indicating that necrotic enteritis infection led to intestinal injury and adversely affected growth performance. However, dietary supplementation with *Lactobacillus plantarum* HW1 mitigated the negative effects in broilers challenged necrotic enteritis. Previous studies found that *Lactobacillus plantarum* alleviated the depressed growth performance and intestinal damage of broilers under *Clostridium perfringens* challenge [16,28]. Moreover, Wang et al. [15] reported that *Lactobacillus plantarum* P8 could decrease the number of fecal oocysts and intestinal lesion scores, and improve growth performance in broilers challenged with *Eimeria*.

The organ indices of the thymus, spleen, and bursa of Fabricius are an important index to evaluate the immune function and health status of broilers [29,30]. In this study, necrotic enteritis infection decreased the thymus index in broilers. This was consistent with a study of Song et al. [31], which also found that necrotic enteritis could decrease the thymus index, and increase peripheral blood NO levels and lysozyme activity in broilers with necrotic enteritis. The above results suggested that necrotic enteritis infection could cause systemic infection in broilers. However, dietary supplementation with *Lactobacillus plantarum* HW1 could increase the thymus index, indicating that *Lactobacillus plantarum* HW1 enhanced the immune system in broilers infected with necrotic enteritis. Zhang et al. [32] found that dietary supplementation with *Lactobacillus acidophilus* in water increased the thymus index and the spleen index in broilers. 

Villus height and crypt depth are important morphological parameters of the intestinal barrier, which play an important role in nutrient digestion and absorption [33,34]. Necrotic enteritis can cause various morphological changes in the gut, such as the decrease in villus height and V/C value [35]. In this study, dietary supplementation with *Lactobacillus plantarum* HW1 could increase the villus height and V/C value of broilers under necrotic enteritic challenge, indicating that dietary supplementation with *Lactobacillus plantarum* HW1 could protect small intestinal villi from the damage of necrotic enteritis, thus promoting the digestion and absorption of intestinal villi and improving the production performance in broilers. Similar results were found in studies by Yin et al. [36] and Wu et al. [37]. 

Tight junction proteins are the key components involved in intestinal barrier protection, and play an important role in blocking the entry of unwanted microorganisms, antigens, and related toxins [38]. In addition, DAO and *D*-lactic acid in the intestinal mucosal epithelium are released into the blood due to intestinal barrier damage [39,40]. Our study found that necrotic enteritis infection significantly decreased the ileal expression of occludin, and decreased the serum DAO and *D*-lactic acid levels in broilers. However, *Lactobacillus plantarum* HW1 could alleviate the adverse effects in necrotic-enteritis-challenged broilers. The results demonstrated that *Lactobacillus plantarum* HW1 alleviated the intestinal damage and improved the intestinal barrier function in necrotic-enteritis-challenged broilers. Other studies also found that *Lactobacillus plantarum* decreased the *D*-lactic acid and DAO levels, and increased the protein expression of occludin in broilers challenged by lipopolysaccharide or under heat stress [41,42].

In response to pathogen stimulation, the intestinal epithelium can generate pro-inflammatory factors like IL-1β, TNF-α, and IFN-γ, which are intimately involved in triggering the inflammatory response within the intestines [43,44]. IL-10, an essential anti-inflammatory cytokine, inhibits pro-inflammatory cytokines including IL-1β and IL-6 [45]. In the present study, necrotic enteritis increased the IL-1β, TNF-α, and IFN-γ levels in the ileum, and decreased the IL-10 level, indicating that necrotic enteritis infection caused the balance between pro-inflammatory and anti-inflammatory cytokines in the intestine to be broken, resulting in intestinal inflammation. However, dietary supplementation with *Lactobacillus plantarum* HW1 could alleviate the adverse effects of necrotic enteritis in broilers. Similar results were found in studies by Deng et al. [46] and Gong et al. [16], who found that *Lactobacillus plantarum* could decrease the levels of IL-1β, TNF-α, and IFN-γ, and increase the IL-10 level in broilers under *Listeria monocytogenes* infection or *Clostridium perfringens* infection. Therefore, this study may provide new evidence for developing *Lactobacillus plantarum* HW1 as a novel feed additive to diet to prevent broilers from inflammatory response.

Gut microflora has important effects on growth performance and immune function by interacting with nutrition utilization and the development of gut system of broilers [47,48]. In our study, necrotic enteritis infection increased the *Clostridium perfringens*, *Salmonella*, *Escherichia coli*, and *Staphylococcus aureus* counts, and decreased the *Lactobacillus* count in the cecum of broilers, suggesting that necrotic enteritis infection caused cecal microflora disorder in broilers. However, *Lactobacillus plantarum* HW1 was able to alleviate the increase in harmful bacteria and improve the *Lactobacillus* count. These results were similar to a previous report in which *Lactobacillus plantarum* B1 decreased the *Escherichia coli* count and increased the *Lactobacillus* count in cecal digesta of broilers [13]. In addition, *Lactobacillus* exerts antibacterial activity by producing short-chain fatty acids and organic acids, thereby reducing the colonization of harmful bacteria, further facilitating the intestinal micro-ecosystem [49,50].

## 5. Conclusions

Dietary supplementation with *Lactobacillus plantarum* HW1 improved growth performance, immune organ indices, intestinal inflammatory response, barrier function, and cecal microflora in broilers with necrotic enteritis. Considering all these indices, *Lactobacillus plantarum* HW1 has potential for use as an alternative to antibiotics for the relief of necrotic enteritis in broiler production.

## Figures and Tables

**Figure 1 animals-13-03810-f001:**
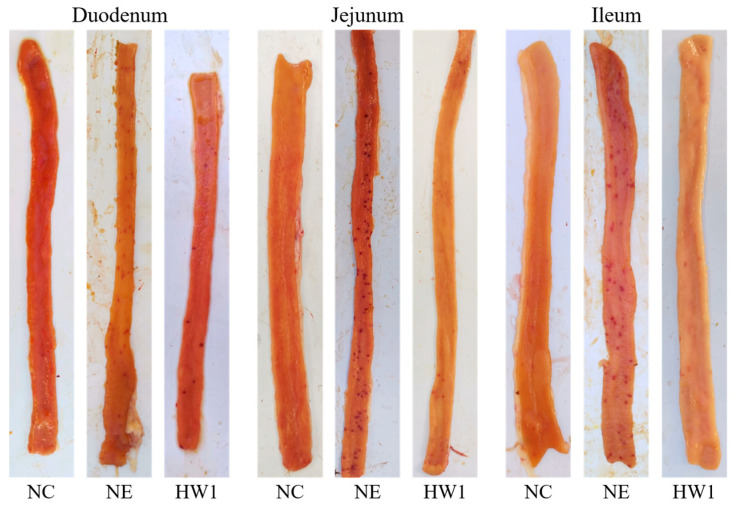
Effect of *Lactobacillus plantarum* HW1 on intestinal lesion scores in broilers with necrotic enteritis (*n* = 6 per experimental group). NC, basal diet; NE, basal diet + necrotic enteritis challenge; HW1, basal diet + 4 × 10^6^ CFU/g *Lactobacillus plantarum* HW1 + necrotic enteritis challenge.

**Figure 2 animals-13-03810-f002:**
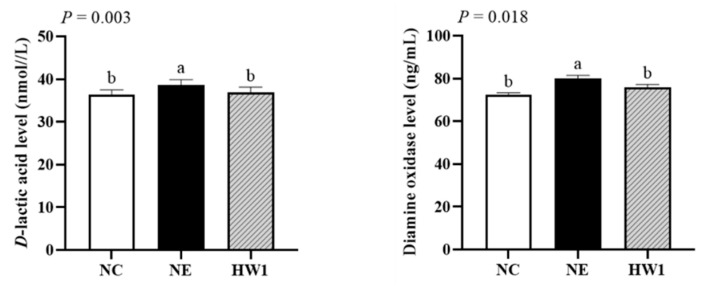
Effect of *Lactobacillus plantarum* HW1 on *D*-lactic acid and diamine oxidase in broilers with necrotic enteritis (*n* = 6 per experimental group). ^a,b^ Different letters indicate significant differences (*p* < 0.05). NC, basal diet; NE, basal diet + necrotic enteritis challenge; HW1, basal diet + 4 × 10^6^ CFU/g *Lactobacillus plantarum* HW1 + necrotic enteritis challenge.

**Figure 3 animals-13-03810-f003:**
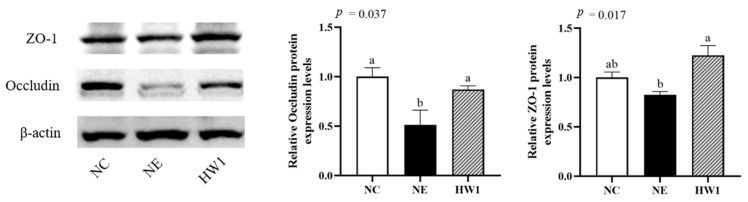
Effect of *Lactobacillus plantarum* HW1 on the tight junction protein expressions in the ileal mucosa of broilers with necrotic enteritis (*n* = 6 per experimental group). ^a,b,ab^ Different letters indicate significant differences (*p* < 0.05). NC, basal diet; NE, basal diet + necrotic enteritis challenge; HW1, basal diet + 4 × 10^6^ CFU/g *Lactobacillus plantarum* HW1 + necrotic enteritis challenge (Appendix A).

**Figure 4 animals-13-03810-f004:**
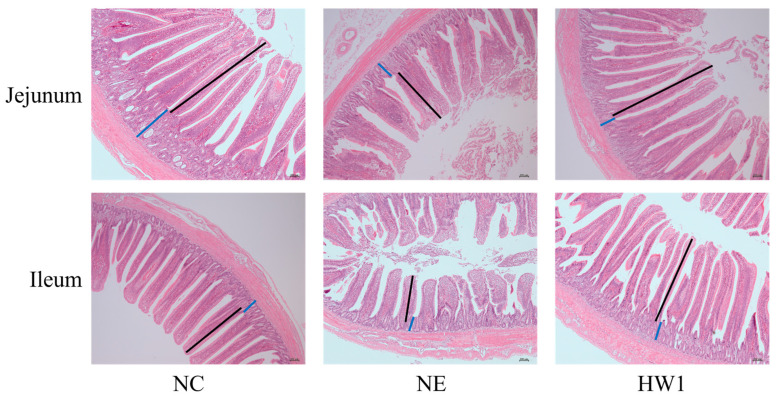
Effect of *Lactobacillus plantarum* HW1 on intestinal morphology in broilers with necrotic enteritis (*n* = 6 per experimental group). NC, basal diet; NE, basal diet + necrotic enteritis challenge; HW1, basal diet + 4 × 10^6^ CFU/g *Lactobacillus plantarum* HW1 + necrotic enteritis challenge. Villus height: black line; Crypt depth: blue line. Images of transverse section were photographed under 50× magnification.

**Table 1 animals-13-03810-t001:** The timetable of the experimental design.

Groups	NC	NE	HW1
Day of the start of the experiment	0	0	0
Number of broilers per group	60	60	60
Day of challenge with *Eimeria* spp.	-	14	14
Days of challenge with *Clostridium perfringens*	-	19–21	19–21
Days of treatment with *Lactobacillus plantarum*	-	-	0–28
Day of the end of the experiment	28	28	28
Number of evaluated broilers per group	6	6	6

**Table 2 animals-13-03810-t002:** Ingredients and nutrient levels in the basal diet (air-dried basis) %.

Item	Contents	
	0–14 Days of Age	15–28 Days of Age
Ingredients		
Corn	59.63	63.96
Soybean meal	34.08	29.55
Soybean oil	2.67	3.26
Limestone	2.00	1.75
CaHPO_4_	0.48	0.30
DL-Methionine	0.41	0.42
NaCl	0.30	0.30
L-lysine HCl	0.15	0.19
L-threonine	0.08	0.07
Premix ^1^	0.10	0.10
Multi-vitamin ^2^	0.10	0.10
Total	100.00	100.00
Total nutrient levels ^3^		
Metabolizable energy (MJ/kg)	12.48	12.78
Crude protein, %	21.74	19.95
Calcium, %	0.98	0.82
Available phosphorus, %	0.60	0.42
Lysine, %	1.28	1.18
Methionine, %	0.51	0.49
Threonine, %	0.88	0.80

^1^ Provided per kilogram of diet: Fe (as ferrous sulfate) 40 mg; Cu (as copper sulfate) 15 mg; Zn (as zinc sulfate) 100 mg; Mn (as manganese sulfate) 100 mg; Se (as sodium selenite) 0.35 mg; I (as potassium iodide) 1.0 mg. ^2^ Provided per kilogram of diet: vitamin A (trans-retinyl acetate) 10,000 IU; vitamin D_3_ (cholecalciferol) 5000 IU; vitamin E (all-rac-α-tocopherol acetate) 80 IU; vitamin K_3_ (menadione) 3.0 mg; vitamin B_1_ (thiamin) 3.0 mg; vitamin B_2_ (riboflavin) 10.0 mg; vitamin B_6_ (pyridoxine HCl) 4.0 mg; vitamin B_12_ (cobalamin) 0.02 mg; nicotinic acid 60 mg; calcium pantothenate 15 mg; folic acid 2.0 mg; biotin 0.15 mg. ^3^ The nutrient levels were calculated values.

**Table 3 animals-13-03810-t003:** Effect of *Lactobacillus plantarum* HW1 on growth performance in broilers with necrotic enteritis from days 0 to 28 (*n* = 6 per experimental group).

Items ^1^	NC	NE	HW1	SEM ^2^	*p*-Value
Day 0 to 14					
Average daily gain (g)	33.75	31.84	33.46	0.552	0.330
Average daily feed intake (g)	43.11	42.42	41.99	0.241	0.157
Feed/gain ratio	1.28	1.34	1.26	0.023	0.317
Day 15 to 28					
Average daily gain (g)	64.18 ^a^	56.97 ^b^	63.47 ^a^	1.269	0.026
Average daily feed intake (g)	96.54 ^a^	90.04 ^b^	94.30 ^ab^	1.087	0.035
Feed/gain ratio	1.51	1.59	1.49	0.030	0.413
Day 0 to 28					
Average daily gain (g)	48.85 ^a^	43.40 ^b^	48.46 ^a^	0.904	0.012
Average daily feed intake (g)	70.49 ^a^	66.39 ^b^	68.45 ^ab^	0.629	0.013
Feed/gain ratio	1.45	1.54	1.42	0.026	0.142
Mortality (%)	0.00 ^b^	8.33 ^a^	1.67 ^a^	2.415	0.031

^a,b,ab^ Different letters indicate significant differences (*p* < 0.05). ^1^ NC, basal diet; NE, basal diet + necrotic enteritis challenge; HW1, basal diet + 4 × 10^6^ CFU/g *Lactobacillus plantarum* HW1 + necrotic enteritis challenge. ^2^ SEM, standard error of the mean.

**Table 4 animals-13-03810-t004:** Effect of *Lactobacillus plantarum* HW1 on intestinal lesion scores and oocysts per gram of excreta in broilers with necrotic enteritis (*n* = 6 per experimental group).

Items ^1^	NC	NE	HW1	SEM ^2^	*p*-Value
Oocysts per gram of excreta ^3^ (×10^6^/g of excreta)	0.00 ^a^	2.87 ^c^	0.66 ^b^	3.023	<0.001
Duodenum lesion score ^4^	0.00 ^b^	2.50 ^a^	0.83 ^b^	2.946	<0.001
Jejunum lesion score ^4^	0.00 ^b^	2.67 ^a^	1.67 ^b^	2.895	0.001
Ileum lesion score ^4^	0.00 ^c^	3.00 ^a^	1.67 ^b^	2.975	<0.001

^a,b,c^ Different letters indicate significant differences (*p* < 0.05). ^1^ NC, basal diet; NE, basal diet + necrotic enteritis challenge; HW1, basal diet + 4 × 10^6^ CFU/g *Lactobacillus plantarum* HW1 + necrotic enteritis challenge. ^2^ SEM, standard error of the mean. ^3^ The oocytes of excreta are observed and counted by microscope. ^4^ A score of 0 (no gross lesions), 1 (congested intestinal mucosa), 2 (small focal necrosis or ulceration; 1 to 5 foci), 3 (focal necrosis or ulceration; 6 to 15 foci), or 4 (focal necrosis or ulceration; 16 or more foci).

**Table 5 animals-13-03810-t005:** Effect of *Lactobacillus plantarum* HW1 on the values for immune organ indices in broilers with necrotic enteritis (*n* = 6 per experimental group).

Items ^1^	NC	NE	HW1	SEM ^2^	*p*-Value
Thymus index (g/kg)	3.40 ^a^	2.96 ^b^	3.69 ^a^	0.099	0.003
Spleen index (g/kg)	1.31 ^a^	1.06 ^b^	1.19 ^ab^	0.044	0.045
Bursa of Fabricius index (g/kg)	2.28	2.06	2.23	0.067	0.415

^a,b,ab^ Different letters indicate significant differences (*p* < 0.05). ^1^ NC, basal diet; NE, basal diet + necrotic enteritis challenge; HW1, basal diet + 4 × 10^6^ CFU/g *Lactobacillus plantarum* HW1 + necrotic enteritis challenge. ^2^ SEM, standard error of the mean.

**Table 6 animals-13-03810-t006:** Effect of *Lactobacillus plantarum* HW1 on the immune-related indices in the ileal mucosa of broilers with necrotic enteritis (*n* = 6 per experimental group).

Items ^1^	NC	NE	HW1	SEM ^2^	*p*-Value
Interleukin 1β (pg/mg)	7.63 ^ab^	8.07 ^a^	7.48 ^b^	0.105	0.047
Interleukin 6 (pg/mg)	2.14	2.22	2.13	0.036	0.598
Interleukin 10 (pg/mg)	15.15 ^a^	13.60 ^b^	15.10 ^a^	0.266	0.015
Tumor necrosis factor α (pg/mg)	5.50 ^b^	5.94 ^a^	5.57 ^b^	0.077	0.035
Interferon γ (pg/mg)	36.81 ^b^	40.95 ^a^	39.36 ^ab^	0.698	0.039
Secretory immunoglobulin A (μg/mg)	1.62	1.54	1.61	0.021	0.271

^a,b,ab^ Different letters indicate significant differences (*p* < 0.05). ^1^ NC, basal diet; NE, basal diet + necrotic enteritis challenge; HW1, basal diet + 4 × 10^6^ CFU/g *Lactobacillus plantarum* HW1 + necrotic enteritis challenge. ^2^ SEM, standard error of the mean.

**Table 7 animals-13-03810-t007:** Effect of *Lactobacillus plantarum* HW1 on intestinal morphology in broilers with necrotic enteritis (*n* = 6 per experimental group).

Items ^1^	NC	NE	HW1	SEM ^2^	*p*-Value
Jejunum					
Villus height (µm)	1147.23 ^a^	1069.55 ^b^	1204.25 ^a^	19.790	0.008
Crypt depth (µm)	230.60	225.32	200.87	6.777	0.163
Villus height/crypt depth value	5.05 ^b^	4.82 ^b^	5.99 ^a^	0.201	0.025
Ileum					
Villus height (µm)	860.43 ^a^	773.22 ^b^	830.04 ^b^	13.147	0.009
Crypt depth (µm)	170.26	170.17	172.26	2.825	0.936
Villus height/crypt depth value	5.05	4.56	4.83	0.093	0.085

^a,b^ Different letters indicate significant differences (*p* < 0.05). ^1^ NC, basal diet; NE, basal diet + necrotic enteritis challenge; HW1, basal diet + 4 × 10^6^ CFU/g *Lactobacillus plantarum* HW1 + necrotic enteritis challenge. ^2^ SEM, standard error of the mean.

**Table 8 animals-13-03810-t008:** Effect of *Lactobacillus plantarum* HW1 on bacterial counts in the cecum of broilers with necrotic enteritis (*n* = 6 per experimental group).

Items ^1^	NC	NE	HW1	SEM ^2^	*p*-Value
Total plate count (log_10_ CFU/g)	7.83	7.99	7.88	0.087	0.777
*Lactobacillus* (log_10_ CFU/g)	6.84 ^a^	6.19 ^b^	6.68 ^a^	0.081	<0.001
*Clostridium perfringens* (log_10_ CFU/g)	5.34 ^c^	6.25 ^a^	5.81 ^b^	0.117	<0.001
*Salmonella* (log_10_ CFU/g)	4.09 ^b^	5.17 ^a^	3.96 ^b^	0.167	<0.001
*Escherichia coli* (log_10_ CFU/g)	4.27 ^a^	4.66 ^a^	3.52 ^b^	0.151	0.001
*Staphylococcus aureus* (log_10_ CFU/g)	6.39 ^b^	6.59 ^a^	6.30 ^b^	0.048	0.020

^a,b,c^ Different letters indicate significant differences (*p* < 0.05). ^1^ NC, basal diet; NE, basal diet + necrotic enteritis challenge; HW1, basal diet + 4 × 10^6^ CFU/g *Lactobacillus plantarum* HW1 + necrotic enteritis challenge. ^2^ SEM, standard error of the mean.

## Data Availability

All data sets collected and analyzed in this study are available at the request of the corresponding authors.

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
