# Peer review of "Effects of Lactobacillus plantarum HW1 on Growth Performance, Intestinal Immune Response, Barrier Function, and Cecal Microflora of Broilers with Necrotic Enteritis"

_animals, 2023, doi:10.3390/ani13243810_

Round 1
Reviewer 1 Report
Comments and Suggestions for Authors
Dear authors
The manuscript “Effects of Lactobacillus plantarum HW1 on growth performance, intestinal immune response, barrier function, and cecal microflora of broilers with necrotic enteritis” is well written, the description of the methodology is adequate and the results shown are compatible with the methodology used.
I only have a few suggestions for the authors.
1) The authors could provide a better description of the Lactobacillus plantarum HW1 isolate.
2) The authors could inform the reader of the criteria used to define the dose of Lactobacillus plantarum HW1, oocysts of Eimeria ssp, and Clostridium perfringes CVCC 2030 used in the study.
Reviewer 2 Report
Comments and Suggestions for Authors
The manuscript entitled “Effects of Lactobacillus plantarum HW1 on growth performance, intestinal immune response, barrier function, and cecal microflora of broilers with necrotic enteritis”, focuses on the very important problem responsible for the economic loss in the poultry industry. Necrotic enteritis, caused mainly by Clostridium perfringens type A, can be and is prevented and treated by antibiotic therapy, however, the lack of dosage control can lead among others to the induction of multidrug resistance a causative agent. The Authors propose in their studies a new pathway for preventing necrotic enteritis by application of Lactobacillus plantarum to 1-day-old broilers. This is a very interesting and promising concept, since the proper modification of the gastrointestinal tract microbiota may result in the maintenance of the health of poultry flocks.
Unfortunately, the manuscript requires serious revision. The data included in the manuscript should provide information allowing the repetition of the experiment. However, the data included in the manuscript do not allow to obtain the similar data. The Authors focused on the evaluation of the L. plantarum effect on the induced necrotic enteritis. There is no information in the whole manuscript on when and how broilers were treated with this bacterial strain. Was it performed on the hatching day, the day before the challenge with Eimeria spp., the day when C. perfringens was introduced, or the day after, whether L. plantarum was given daily together with drinking water or with the diet? It has to be clarified. Moreover, in the section “Materials and Methods”, the Authors stated that the HW1 group obtained basal diet + 4x106 CFU/g L. plantarum. The question is whether the number of L. plantarum was calculated per gram of diet or gram of the broiler. How the proper number of bacteria was delivered to each broiler and how the intake of L. plantarum per individual was controlled? This section requires revision because it has an impact on the analysis and reliability of obtained data.
The Authors should introduce to their manuscript the timetable which will clearly state the day of the start of the experiment, the division into experimental groups including the number of broilers, the day of challenge with Eimeria spp. and days of challenge with C. perfringens, treatment with L. plantarum, then the day of the end of the experiment including the number of evaluated animals. It is very important since the indicated number of broilers does not correspond with the information provided in the section “2.5. Sample collection”. In this section, the Authors stated that “on day 28, one broiler from each replicate was randomly selected”, and that makes 18 broilers. The result section indicates the number of analyzed broilers = 6. Maybe the Authors meant 6 broilers per group, but it should be clearly stated. The sequence of described events should be the first collection of the blood, then killing birds by cervical dislocation. Please, provide information on what coagulant was used in the vacuum tubes.
In the section “2.6. Measurement of intestinal….” the Authors described the evaluation of the excreta on day 19. In the next sentence (line 136), the Authors stated “All the sampled broilers were scored for the intestinal lesions in the duodenum, jejunum, and ileum by visual examination according to the method of Johnson et al.” Could you clarify whether this sentence refers to broilers killed on the 28th day or assessed on the 19th day? In section 3.2., the Authors have to include the images of the representative scores of the intestinal lesions.
The Authors stated that the chosen cytokines and sIgA were evaluated in the ileal mucosa. First, there is no description of how ileal mucosa was prepared for the evaluation, and it has to be included regarding the reproducibility of the experiment. Second, the Authors in section 2.8 indicated that the following cytokines were assessed: IL-1b, IL-8 (should be CXCL8), IL-10, IL-17, TNF-a, IFN-g. However, in section 3.4. the Authors present results regarding the following cytokines: IL-1b, IL-10, TNF-a, IFN-g. There is no information regarding CXCL8 and IL-17, but suddenly IL-6 is present.
In section 2.9. line 151, the information regarding when the samples were collected should be added. One can guess that the collection was performed on the 28th day but it has to be clearly said.
Section 2.11, lines 174-175, please, revise the sentence, since in this form, it suggests that only one gram of the cecal content was collected per one group.
Lines: 214, 230240, 252, 261, 290 the phrase “NE treatment” should be changed. Necrotic enteritis was induced by the introduction of Eimeria spp., and C. perfringens. Therefore, it should be changed to the phrase “NE induction” or “NE stimulation”
Table nr 2, 3, 4, 5, and 6, please, change the titles. The broilers cannot be challenged with a disease, which means necrotic enteritis. In broilers, necrotic enteritis was induced.
Table 7. The results in the table do not correspond with the description regarding the Lactobacillus count in the HW1 group.
Lines 292 and 295; please, change the word; the word “deceased” refers to people
In each table, n=6 should be changed to n=6 per experimental group.
Line 53: the word “products” should be changed to “produces”
Although the concept of the studies is very interesting, the errors in the presented manuscript require deep correction. Without the proper revision, I will not recommend the manuscript for publication
Comments on the Quality of English Language
The language of the manuscript requires moderate correction
Round 2
Reviewer 2 Report
Comments and Suggestions for Authors
The revised version of the manuscript entitled “Effects of Lactobacillus plantarum HW1 on growth performance, intestinal immune response, barrier function, and cecal microflora of broilers with necrotic enteritis” has highly improved and the Authors met the majority of the issues. However, still, there is one error. The Authors in the first version of the manuscript stated that they collected blood using tubes with COAGULANT. Because of this statement, they were required to provide information regarding this compound. The Authors in the revised version stated that blood samples were collected in the tubes with heparin sulfate. This is anti-coagulant therefore plasma samples were obtained after centrifugation not sera. Please, correct this throughout the manuscript.
After revision, I will recommend the manuscript for publication.
